# An Update on Polyphosphate In Vivo Activities

**DOI:** 10.3390/biom14080937

**Published:** 2024-08-02

**Authors:** Robert Schoeppe, Moritz Waldmann, Henning J. Jessen, Thomas Renné

**Affiliations:** 1Institute of Clinical Chemistry and Laboratory Medicine (O26), University Medical Center Hamburg-Eppendorf, D-20246 Hamburg, Germany; 2Institute of Organic Chemistry, Albert-Ludwigs-University of Freiburg, D-79105 Freiburg, Germany; henning.jessen@oc.uni-freiburg.de; 3Irish Centre for Vascular Biology, School of Pharmacy and Biomolecular Sciences, Royal College of Surgeons in Ireland, D02 YN77 Dublin, Ireland; 4Center for Thrombosis and Haemostasis (CTH), Johannes Gutenberg University Medical Center, D-55131 Mainz, Germany

**Keywords:** polyphosphate, energy metabolism, exopolyphosphatase, contact activation

## Abstract

Polyphosphate (polyP) is an evolutionary ancient inorganic molecule widespread in biology, exerting a broad range of biological activities. The intracellular polymer serves as an energy storage pool and phosphate/calcium ion reservoir with implications for basal cellular functions. Metabolisms of the polymer are well understood in procaryotes and unicellular eukaryotic cells. However, functions, regulation, and association with disease states of the polymer in higher eukaryotic species such as mammalians are just beginning to emerge. The review summarises our current understanding of polyP metabolism, the polymer’s functions, and methods for polyP analysis. In-depth knowledge of the pathways that control polyP turnover will open future perspectives for selective targeting of the polymer.

## 1. Introduction

Inorganic polyphosphate (polyP) is a linear anionic polymer composed of three to several hundred orthophosphate (P_i_) residues that are linked by energy-rich phosphoanhydride bonds [1,2]. In microorganisms, polyP has a variety of cellular functions, including phosphate and energy storage, support of survival in the stationary phase and under nutrient starvation, and, together with polyphosphate kinases, regulation of cell motility, biofilm formation, and virulence [3,4]. Furthermore, the degradation of polyP reduces stress-induced DNA damage in microorganisms and regulates their enzyme activity, which has implications for virulence and sensitivity for antibiotics. In yeast, polyP is crucial for controlling P_i_ homeostasis but also participates in adaptive processes, including growth and heavy metal ion tolerance [2]. PolyP constitutes a metabolic energy-storing compound in eukaryotes and fuels an array of cellular processes [5,6]. Polyphosphate levels are affected by many pathways in different cells. However, in contrast to unicellular eukaryotes and prokaryotes, little is known about enzymes that regulate polyP in mammalians.

## 2. Biophysical Characteristics of Polyphosphate

Polyphosphate is an inorganic anionic energy-rich polymer widespread in all living organisms, from bacteria to mammalians [7]. Furthermore, the polymer is also produced synthetically, and synthesised polyP is used in large amounts in an array of applications, e.g., as a water softener, fire extinguisher, or as additives for conservation, pH buffer, and water retention in the food industry [8]. Synthetic polyP molecules are technically produced by condensation of orthophosphoric acid monomers with dehydration under high temperatures. Polymerisation generates energy-rich phosphoanhydrid bonds that form the backbone of linear and branched polymers and supramolecular polyP structures [9]. The nucleotides GTP and ATP share these high-energy phosphoanhydride bonds with polyP. In nature, enzymatic hydrolysis of the terminal high-energy β–γ and α–β phosphoanhydride bonds in ATP to ADP and ADP to AMP, respectively, liberates ~30 kJ/mol of energy [5]. Corresponding to the hydrolysis of the terminal ATP phosphoanhydride bond, the hydrolysis of P-O-P anhydride bonds in linear polyP similarly generates ~30 kJ/mol under standard conditions [10,11]. The kinetics of the P_i_-generating hydrolysis reaction is slow. At physiological pH, polyP has a half-life of about 90 min in plasma [12] and several months in aqueous solutions ex vivo. However, it vastly decreases upon increasing temperature and decreasing pH [1,13]. Synthetic polyP molecules are linear, cyclic, or branched and have a chain length from three up to several thousand orthophosphate units [14]. Branched polyP, so-called ultraphosphates, have not been found in living organisms [15]. Their stability depends on pH and the presence or absence of divalent metal cations [16,17]. It is currently unclear whether they are not formed in biological systems or are not found due to their short half-life in aqueous systems, as the branching sites are rapidly hydrolysed [17]. Cyclic polyPs (i.e., metaphosphates) have been synthesised from linear polyP upon solvent removal, and they are comparably stable in aqueous solutions. The linear polyP backbone has a high degree of conformational flexibility with staggered and eclipsed conformations dependent on whether monovalent (e.g., NH_4_^+^, Na^+^, or K^+^), divalent (e.g., Mg^2+^, Zn^2+^, Ca^2+^), or even trivalent (e.g., Gd^3+^) cations are complexed to the phosphate backbone [18,19]. Molecular modelling and experimental data revealed that binding Ca^2+^ ions to polyP molecules leads to aggregate/microparticle formation ex vivo [20]. The length of a polyP molecule in its linear conformation can be estimated from the size of a single phosphate monomer (2.7 Å), indicating that long-chain polyP polymers (around 1000 P_i_ units) of bacterial and mammalian cell origin is up to ∼300 nm in length. However, the natural polymer is complexed to divalent cations and proteins that compact the polyP within subcellular compartments such as acidocalcisomes or platelet-dense granules. The phosphates in the polyP backbone are strongly acidic (pK_a_ ∼0–3) molecules and are fully deprotonated at physiological pH [21,22]. The negatively charged oxygen atoms from two adjacent phosphate groups complex divalent cations. Thus, polyP provides a scaffold for the local enrichment of Zn^2+^ in platelet-dense granules and Ca^2+^ in various intracellular compartments [23]. The complexed cations, at least in part, neutralise the polyanionic polymer backbone [24,25,26] (Figure 1).

PolyP in living cells are linear polymers characterised by their mean chain length, distribution of chain length (dispersity), and binding partners, including cations and proteins. PolyP is ubiquitously found in higher organisms, including all mammalian species [28]. However, a comprehensive overview of the organ-specific polymer content and chain length does not currently exist. This is urgently needed to provide the basis for a rational assessment of the (patho)physiological functions of the polymer. Classical studies in rats and mice indicate that polyP content is high in the heart and brain, exceeding polymer levels in the liver by three-fold [5,29]. Based on the high affinity of polyP for binding to Ca^2+^ (K_D_ in the lower nM range) that exceeds extra- (2.2–2.6 mM) and intracellular (0.1–1.0 mM) Ca^2+^ levels, the natural polymer is saturated with Ca^2+^ ions. Ca^2+^-binding essentially regulates the biophysical properties of the molecule and makes extracellular polyP virtually insoluble in biological fluids [30,31]. Synthetic polyP complexed with divalent cations (Ba^2+^, Pb^2+^, Mg^2+^, or Ca^2+^) is either insoluble or dissolves in tiny amounts only in aqueous solutions. In contrast, short-chain synthetic polyP complexed with Na^+^, K^+^, or NH_4_^+^ ions is soluble in water and used as a water softener, e.g., in dishwasher tablets [9,32,33,34]. Both short- and long-chain polymers are virtually insoluble in ethanol, which can be used to precipitate the polymer from biological sources [9,32,33,34]. Precise data on the affinity of polyP for the various complexed cations is not available. Because polyP degrades over time, the total charge of the molecule and binding affinities for multiple cations have remained undefined. Furthermore, no detailed data exist on natural polyP secondary structures, e.g., on cell surfaces or with internal storage pools. However, the fact that intercalating dyes such as DAPI or Sytox Green/Orange stain polyP suggests that at least parts of the polymer have an α-helix-rich structure (reviewed in [35]) similar to DNA [36].

## 3. PolyP Metabolism

It has been established for more than 60 years that polyP is produced in all living organisms [37,38]; nevertheless, little is known about the metabolism, regulation, concentration, chain length, and localisation of the polymer in mammalian cells. There seem to be significant differences between various cell types that most likely reflect diverse but distinct functions of the polymer in subcellular compartments, which are described below and summarised in Table 1.

## 4. Bacteria

In bacteria, polyP is synthesised and degraded by polyphosphate kinases (PPKs). Two distinct enzymes, PPK1 and PPK2 that differ in their reaction profile [41], reversibly catalyse the polymerisation of the terminal phosphate of ATP into a nascent polyP chain or transfer terminal Pi from the polymer backbone to GDP, synthesising GTP [11,58]. PPK2 is an 80 kDa enzyme bound to the inner leaflet of the plasma membrane and is evolutionarily older than PPK1 [59]. By modulating cellular energy status, polyP protects bacteria from various external stressors, including heat, ultraviolet irradiation, antibiotics, and oxidative stress. PPK1 deficiency by target ablation of ppk1 gene expression impairs growth and biofilm formation and reduces resistance to starvation [60,61]. Moreover, gene ablation of *ppk2* in *Mycobacterium smegmatis* leads to biofilm synthesis and virulence [62,63]. *Corynebacterium glutamicum* expressed two PPK2 isoenzymes, while *Pseudomonas aeruginosa* and *Sinorhizobium meliloti* have three isoenzymes each, and *Ralstonia eutropha* even expresses five PPK2 variants [64]. Class I PPK2´s phosphorylate nucleoside diphosphates ADP and GDP by transferring the terminal P_i_ of polyP to the nucleotide diphosphates and vice versa, shuttling P_i_ to the nascent polyP chain [65]. In contrast, class II PPK2´s catalyse nucleotide monophosphate phosphorylation and were previously called polyP-AMP phosphotransferase (PAP) [66]. Class III PPK2´s phosphorylate mono- or diphosphates in nucleotide to generate ADP or ATP, respectively, using polyP as a phosphate donor [67].

Exopolyphosphatases (PPXs) hydrolyse polyP of chain length > 3 to release the terminal P_i_. Based on the primary structure, they are classified into two types, PPX1 and PPX2, respectively. Both enzymes are members of the same protein family, the PPX-GppA phosphatases. Most bacteria, such as *Corynebacterium glutamicum*, express two distinct exopolyphosphatases. PPX1 and PPX2 from *C. glutamicum* share low identity on the protein level (25%) and are different from enterobacterial, archaeal, and yeast exopolyphosphatases [57]. X-ray studies of the crystal structure of *E. coli* PPX1 at 2.2-°A resolution revealed that the molecule is composed of four domains, with two of them mediating binding to polyP > 35-P_i_ chain length and two harbouring the enzymatic activity, respectively. Bacterial PPX enzymes require both Mg^2+^ and K^+^ for maximal activity [68].

## 5. Unicellular Eukariotes

PolyP presence is evolutionary and not restricted to prokaryotes. The slime mould *Dictyostelium discoideum* and various fungi, such as the model system Saccharomyces cerevisiae, contain polyP. Eukaryotes express a polyP-modifying enzyme not found in bacteria, endopolyphosphatase1 (PPN1). PPN1 has endo- but also exopolyphosphatase activities and is found in various compartments, including the cytosol, the nuclear and mitochondrial membranes, and within vacuoles [69,70,71]. At these sites, PPN1 colocalises with PPX1, and the activity of both polyP-modifying enzymes is significantly increased by Mg^2+^ but not by Ca^2+^ ions [7,25]. The optimal pH for both enzymes is close to 7.0. Similarly to bacterial PPK, the anionic polysaccharide heparin is a competitive inhibitor for yeast endo- and exopolyphosphatases, most likely by competing for the polyanion-binding site in the enzymes [47,72]. Deficient yeast strains in PPX1 and PPN1 still have endopolyphosphatase activity, indicating that other yet unknown enzymes with overlapping substrate specificity exist [69]. Indeed, a yeast strain with vacuole-targeted exopolyphosphatase Ppx1 and concomitantly depletion of the two endopolyphosphatases (*ppn1*Δ*ppn2*Δ, vt-Ppx1) is deficient in polyPase activity [73]. 

A specific polyP-rich organelle are acidocalcisomes that are membrane-bound vesicular structures initially described in *Trypanosoma crucii*; however, they exist in other unicellular eukaryotes, bacteria, and human cells [74]. In addition to polyP, acidocalcisomes are rich in Ca^2+^ and play a role in intracellular calcium signalling and osmoregulation. Acidocalcisomes are acidic, and their specific milieu is maintained by two proton pumps, i.e., vacuolar H^+^-pyrophosphatase (V-H^+^-PPase) and H^+^-ATPase (V-H^+^-ATPase) and a Ca^2+^-ATPase, which mediates the transport of H^+^ into the acidocalcisome lumen in exchange for Ca^2+^. Ca^2+^ ions are released from acidocalcisomes into the cytoplasm via an inositol 1,4,5-trisphosphate receptor (IP_3_ receptor) bound Ca^2+^ channel [75,76,77]. At the acidocalcisome membrane, the vacuolar transporter chaperone complex (VTC) synthesises polyP from ATP and transports the nascent polymer chain into the organelle lumen [42]. The acidocalcisome polyP is hydrolysed by PPX, and the cleaved P_i_ is exported via Na^+^/P_i_ symporters [78]. Acidocalcisomes contain additional polyP-modifying enzymes such as vacuolar soluble PPase (VSP), PPX, and acid phosphatase. Upon hyperosmotic stress, *Trypanosoma crucii* produces long-chain polyP, thus lowering P_i_ and cation levels and consequently reducing osmotic pressure [22]. Based on these data, osmoregulation is considered an essential function of polyP in bacteria and unicellular eukaryotes. However, it is currently unknown whether this fundamental cellular function is conserved in eukaryotes, especially as the intracellular polyP levels are much lower in complex organisms. 

## 6. Mammalians

Mammalian cells contain polyP, although the cellular content of the polymer, its subcellular localisation, and chain length greatly vary [1]. Using [^32^P]-radiolabeled polymers has shown that polyP content in rat hepatocytes is highest in the nucleus (89 ± 7 µM) followed by plasma membranes (43 ± 3 µM), cytosol (12 ± 2 µM), mitochondria (11 ± 0.6 µM) and lowest in microsomes (4 ± 0.5 µM) [5]. High polyP levels in the cytosol have also been found in various immortalised cultured cell lines, including NIH3T3 fibroblasts, Vero epithelial kidney, and Jurkat CD4+ T cells [5]. In contrast to simpler organisms where polyP metabolising enzymes have been well characterised, the polyP synthesising and degrading machinery in mammalians have remained enigmatic. Surprisingly, and despite the fundamental functions of the polymer, the polyp-metabolising enzymes of simpler organisms are not conserved in mammalians, and knowledge about enzymes involved in polyP turnover is just beginning to emerge [79]. Mammalian phospholipase D shares similarities with bacterial PPK1 and PPK2 but has no polyP-modifying activity [7]. Bacterial PPX1 and human (h)-Prune are functionally equivalent but non-homologous proteins originating from distinct ancestors. PPX1 is a member of the PPX-GppA phosphatases family. At the same time, h-Prune belongs to the DHH phosphoesterases superfamily, which shares the phosphodiesterase activity to hydrolyses short-chain polyP (up to 4 P_i_). In contrast to PPX, h-Prune processes long-chain molecules with >25 P_i_ residues poorly. Consistently, recombinant h-Prune is unable to hydrolyse short- and medium-chain polyP. Conversely, the knockdown of h-Prune expression significantly decreases polyP and ATP content in treated HEK293 cells [52]. Prune is involved in mammalian polyP metabolism, although its exact role and regulation have not been fully elucidated [52,80]. Prune is expressed and functions in polyP metabolism in *Drosophila melanogaster*, suggesting that some fundamental principles of polyP regulation are conserved throughout evolution. However, the cellular localisation of Prune differs among the species. In *Drosophila melanogaster*, Prune is predominantly in mitochondria. In contrast, in addition to mitochondria, the polymer is enriched in various other compartments, including ER, cytoplasm, nucleus, and many (secretory) granules and vesicles in mammalian cells [80]. Throughout the species, polyP is enriched in mitochondria and couples phosphate- with ATP-homeostasis. In mammalians, the F0F1-ATP synthase, which generates ATP within the mitochondrial matrix, modulates polyP content and chain length [80,81].

Another group of conserved polyp-metabolising enzymes are alkaline phosphatases (Table 1), ubiquitous membrane-bound glycoproteins that are capable of cleaving phosphoester bonds and additionally have the capacity for degrading polyP [82]. Shrimp alkaline phosphatase efficiently degrades extracellular polyP in plasma, confirming that the enzyme has polyp-metabolising activities [60]. Alkaline phosphatases are similar to yeast cytosolic and nuclear diphosphoinositol polyphosphate phosphohydrolase 1 and 2 (DDP1 and DDP2) [83].

In addition to polyP synthesis and degradation, the polymer is also regulated by the availability of P_i_. The Xenotropic and polytropic retrovirus receptor 1 (XPR1) was initially identified as a docking receptor for retroviruses. However, later studies showed that XPR1 also exports P_i_ in human and murine platelets [84,85]. XPR1 shares homology with Pho84, Pho87, Pho90, and Pho89 P_i_ transporters in *Saccharomyces cerevisiae* [86]. XPR1 is an atypical G-protein-coupled receptor expressed in intracellular organelles and at the plasma membrane. In addition to being regulated by XPR1, polyP itself modulates membrane channels. Primary rat hepatocytes express the mitochondria permeability transition pore (mPTP) at their inner mitochondrial membrane. mPTP is reversibly complexed with the transient receptor potential melastatin 2 (Trpm2) or transient receptor potential channel subfamily A member 1 (TRPA1) [24,87]. PolyP is involved in mPTP activation in murine cardiac myocytes in a polymer chain length-dependent manner. Depleting the polymer protects against Ca2+-induced mPTP opening, suggesting an autocrine regulation of mammalian Ca^2+^-polyP [88]. mPTP regulates the influx of Ca^2+^ and, thus, causes mitochondrial swelling, which is regularly associated with cell apoptosis [89].

## 7. Extracellular PolyP

PolyP has both intracellular and extracellular activities (Table 2). In bacteria and unicellular eukaryotes, polyP is an intracellular polymer. However, when cells disintegrate, the polymer might be exposed to the extracellular environment. In contrast, some eukaryotic cells, including mast cells, astrocytes, and thrombocytes, can actively release the polymer from their secretory storage organelles [30,90]. 

PolyP release and exposure at the cell surface have been identified in thrombocytes [91]. Thrombocytes contain polyP within their dense granules, which appear dark in electron microscopy due to their high Ca^2+^ and P_i_ content [92]. The dense granules share similarities with acidocalcisomes in unicellular eukaryotes [93]. However, unlike intracellular acidocalcisomes, dense granules fuse with the cell membrane upon activation and release intracellular polyP to the extracellular compartment [92]. Elegant real-time microscopy studies have visualised polyP “secretion” [20].

In contrast to released soluble mediators from dense granules such as ATP, platelet polyP is a mostly insoluble polymer. It is retained on the surface of procoagulant platelets as Ca^2+^/Zn^2+^-rich microparticles [20]. Only small amounts of relatively short-chain polymers of ~70–80 P_i_ residues are soluble and found in the plasma compartment. However, it is uncertain whether these polymers are in a solution or if they stick to extracellular vesicles or exosomes [94,95]. Polyanion-exchanger-based polyP extraction from platelets showed that platelets, similar to other mammalian cells, mostly contain long-chain polyP with chain lengths of several hundreds of units. However, a small amount of total polyP is in the range of ~80 P_i_ units [96]. Isolation of platelet polyP using the phenol/chloroform-based extraction method selects for these soluble polymers and results in the purification of short-chain ~80 P_i_ long polymers [14]. The regulation of platelet polyP remains to be fully elucidated. Nevertheless, intracellular P_i_ levels have been identified as critical regulators of cellular polymer content [97], and the conditional ablation of platelet XPR1 increased polyP levels and augmented platelet-driven thrombosis with no bleeding abnormalities [85].

Synthetic and cell-purified polyP has modulated an array of reactions in plasma ex vivo [98,99]. However, the in vivo activities of the polymer seem to be mediated chiefly, if not exclusively, via the factor (FXII)-activated plasma contact system [100]. FXII initiates the plasma contact system that drives the formation of bradykinin (BK) via the kallikrein–kinin system and the clots via the intrinsic coagulation pathway, respectively [101]. Binding to negatively charged polyP microparticles on cell surfaces induces a conformational change in FXII zymogen, resulting in an active serine protease, activated FXII (FXIIa) [102,103]. Supporting a direct polyP-mediated FXII activation, immunofluorescent microscopy colocalised FXIIa and polyP on surfaces of cancer cells and cancer-derived exosomes [90]. Injection of polyP into the skin of normal mice induced BK-mediated oedema. In contrast, mice with inherited deficiency in or pharmacologically inhibited expression of FXII and kinin B2 receptors were protected from polyP-induced vascular leakage, indicating that polyP activates FXIIa that drives BK-stimulated vascular leakage [98,104]. Recently, targeting FXIIa has proven to be an efficient therapy for patients with hereditary angioedema (HAE, a life-threatening swelling disease) [103]. An engineered fully human antibody, Garadacimab^TM^, specifically neutralised FXIIa (originally named 3F7) and potently blocks BK-driven swelling attacks in a prophylactic setting [105]. The exact trigger for FXII activation and kallikrein–kinin-system-mediated BK formation in HAE is not entirely clear; however, clearly, mast cell-derived heparin [106] or polyP (which is a component of mast cell granules [77]) are promising candidates. 

Activated platelets and platelet microparticles stimulate thrombin generation in an FXII-contact activation-dependent manner in vivo and human plasma with critical implications for thrombosis in vivo [98,107]. Vice versa, FXII-dependent plasma clotting and thrombus formation are defective in mice and humans with a genetic deficiency in polyP (inositol hexakisphosphate kinase 1 null [*Ip6k1*^−*/*−^] mice [108] and Hermansky–Pudlak syndrome patient platelets [98], respectively). Natural polyP is largely insoluble Ca^2+^, and Zn^2+^-rich polyP nanoparticles potently initiate FXII contact activation similarly to kaolin (a silicate commonly used to trigger FXII activation in diagnostic aPTT clotting assays) [94]. 

Based on the original discovery that FXII-deficient mice are protected from arterial and venous thrombosis but do not bleed excessively [107,109], the FXIIa-triggered contact system has emerged as a promising target for safe anticoagulant drugs. Mimicking FXII/FXIIa inhibitors targeting extracellular polyP have emerged as a novel treatment strategy to dampen thrombosis with preserved haemostasis [103]. Various polycation agents have been used to target extracellular polyP, including spermidine, histone H1, polymyxin B, and cationic polymers [30]. Neutralising the polyP charge interferes with the ability of the polyanion to initiate FXII contact activation and ablates the prothrombotic activities of the polymer while sparing haemostasis [30,103]. Another strategy for interference with polyP prothrombotic activity is based on polymer backbone degradation. Originally, alkaline phosphatase has been shown to interfere with polyP coagulation activity and to block activated platelet-driven thrombosis in mouse models [110,111]. Alkaline phosphatase cleaves the polymer chain; however, it also removes phosphate groups from lipids and proteins, suggesting the use of polyP-specific phosphatases. *E. coli* exopolyphosphatase (PPX), a member of the PPX-GppA phosphatases family, specifically cleaves polyP of chain length > 35 P_i_ but no other polyanions such as heparin, DNA/RNA, or synthetic dextran sulphate. PPX efficiently degrades synthetic and natural polyP and ablates the procoagulant activity of the polymer in human plasma. Infusion of PPX into mice confers protection against polyP procoagulant activity and blunts activated platelet-stimulated vascular occlusions in mouse thrombosis models [112,113,114]. Similarly to full-size PPX, the recombinant polyP-binding domains of the enzyme (domains 3 and 4) bind with high affinity to polyP, interfere with procoagulant activities of the polymer, and block polyP-mediated FXII contact activation. Both polyP-degrading PPX and non-degrading PPX_∆12 inhibit thrombus formation in lethal pulmonary embolism without interfering with haemostasis in vivo, reproducing the effects of FXII inhibitors [68,115]. The data show that targeting polyP mimics the selective importance of FXIIa in thrombosis, sparing hemostasis [103,116]. The findings with polyP/FXII also suggest that pharmacologic interference with polyP offers the unique opportunity for blocking thrombosis and inflammation (thrombo-inflammation) without affecting physiologic haemostasis at vascular injury sites (safe anticoagulants) [90,117]. The concept of interference with polyP-triggered coagulation seems to have arisen in blood-sucking insects. Sand flies express proteins in their saliva that allow feeding mammalian blood without triggering brad–kinin-mediated itching and FXIIa-mediated clotting. These proteins, PdSP15a and PdSP15b expose a positively charged helix that binds to the polymer and neutralises polyP procoagulant and proinflammatory activities [118]. 

The blood circulating histidine-rich glycoprotein (HRG) binds to polyanions such as DNA, RNA, and polyP. HRG has been shown to inhibit polyP procoagulant activity by interfering with FXII contact activation [119]. The underlying mechanism for antagonising FXII activity is a competition between HRG and FXII for binding to polyP. Reduced FXII binding to polyP results in defective FXII contact activation and aPTT prolongation [120,121]. PolyP injection in HRG-deficient (*Hrg*^−/−^) mice consistently triggers excess fibrin formation [122,123]. Vice versa, the reconstitution of *Hrg*^−/−^ mice with HRG restores excess polyP-driven thrombosis in an FXII-dependent manner [119]. The data are consistent with HRG operating as an endogenous FXII regulator by binding to the contact activator polyP.

In addition to procoagulant platelets and activated mast cells, polyP is released from mammalian astrocytes and has a role in neurotransmission. Astrocyte–polyP activates neurons and provokes calcium-mediated signalling [124]. PolyP is increased in human and mouse astrocytes and cerebrospinal fluid amyotrophic lateral sclerosis (ALS) and frontotemporal dementia (FTD). Targeting secreted polyP in ALS/FTD astrocytes prevents motoneuron death, indicating that polyP released by ALS/FTD astrocytes is a critical factor in neurodegeneration and a potential therapeutic target for interference with the diseases [125]. Intracellular polyP also has chaperone activities in the central nervous system and restores incorrect protein folding, suggesting a role of polyP in amyloidogenic processes and disease states [126,127]. Indeed, polyP improves degenerative neuronal diseases where misfolded protein aggregates seem to mediate the pathology, such as Alzheimer’s, Huntington’s, and Parkinson’s disease [18,128]. Together, the data highlight the importance of polyP in the central nervous system, which is awaiting future studies.

## 8. PolyP Signalling

Established functions of polyP in prokaryotic cells include storage for energy, P_i_ and Ca^2+^, biofilm formation, biomineralisation, and bone formation [2,32,129,130] (Table 2). Importantly, intracellularly, polyP activates mTOR and, thus, promotes cell proliferation and growth [131]. The exopolyphosphatase h-prune provides another link of polyP and mitogenic activity. The binding of h-prune to nm23-h1, a metastasis suppressor, inhibits h-prone phosphatase activity [52]. Defective phosphatase activity, at least in yeast, increases polyP content and might provide a rationale for h-prune expression in cancer associated with poor prognosis via augmented metastasis and tumour growth [52,80]. The precise function of polyP in cancer is likely complex and might be cell-type specific. In myeloma cancer cells, the intracellular polyP level is significantly higher than that of “healthy” primary plasma cells. The addition of synthetic polyP initiates apoptosis of myeloma cells [132]. However, the underlying mechanisms are unclear and might include a change in pH, complexation of surface-bound ions, activation of specific signalling pathways, or local increase in P_i_. In addition to regulating basal cellular functions, it has been suggested that polyP is covalently attached to target proteins [133], e.g., nuclear signal recognition 1 (Nsr1) and its interacting partner, topoisomerase 1 (Top1; polyphosphorylation), on lysine residues within conserved N-terminal polyacidic serine and lysin- (PASK) or histidine-rich clusters [134,135]. However, recent studies have challenged the hypothesis of polyP being a novel noncovalently attached post-translational modification [136,137]. Lysine and histidine residues are positively charged at physiological pH, and these amino acids bind tightly electrostatically (but not covalently) to the negatively charged polyP backbone. The conserved histidine a-helical domain (CHAD) is an example of a polyP-binding protein found in several procaryotic cells [138].

Extracellular polyP has emerged as a procoagulant and proinflammatory mediator. Due to its high biological activity and link to disease states, the polymer has potential diagnostic implications as a (predictive) biomarker with multiple methods for polymer analysis (Table 3). Intercalating dyes can visualise polyP in histology upon separation in agarose and urea gels and individual cells by flow cytometry (FACS) [1]. Specificity for polyP upon UREA-PAGE separation is increased by photobleaching (negative DAPI staining), where polyP appears dark and DNA or proteoglycans turn whitish [36,77,132]. DAPI binds both to polyP and DNA but differs in its emission wavelength. DAPI bound to DNA emits a greenish colour around 461 nm, while DAPI/polyP is bluish at 525 nm [139]. As the dyes bind to helical polyP portions, the polymer backbone needs to exceed >15 P_i_ to allow for detection. Dyes JC-D7 and JC-D8 differ in emission spectrum intensity depending on the polyP chain length with >15 P_i_ [140]. Elegant work using FITC-labelled polyP has shown that cells endocytose polyP microparticles. Using FITC-polyP allows for intracellular polyP staining and the analyses of transport and enrichment of (labelled) polyP to distinct cellular compartments depending on cell activation, metabolic state, and environment [141]. For flow cytometry and histology, neutral red was initially used to visualise polyP [142]. The polyP-binding domain of PPX (PPBD, synonymous to PPX_∆12) selectively binds to polyP with chain length > 38 P_i_ units [142]. PPBD [143] has been used in histology and FACS to visualise and quantify the polymer in an experimental setting and might provide a lead structure for a diagnostic polyP probe with clinical implications. Furthermore, CHADs bind to polyP-rich granules in vivo and may provide an alternative investigation for polyP [144]. Biophysical methods such as small-angle X-ray and ^31^P NMR have been established to analyse the structure of the polymer dependent on bound cations in dry form or solution, respectively. The analysis of polyP in clinical settings, its association with FXII activation states, and its role in diseases requires future research. 

## 9. Future Perspectives

What are the most urgent needs for the field? So far, a comprehensive overview of polyP amount and chain length in all mammalian organs is missing. This “polyP atlas” would provide the basis for analysis of changes in the polymer associated with disease states. Additionally, and as a next step, the amount and composition of polyP in all cellular sub-compartments require attention. Furthermore, the polyp-metabolising enzymes (if existing) need to be identified and fully characterised to allow for selective and specific modulation of polyP content, chain length, and, thus, activities.

## Figures and Tables

**Figure 1 biomolecules-14-00937-f001:**
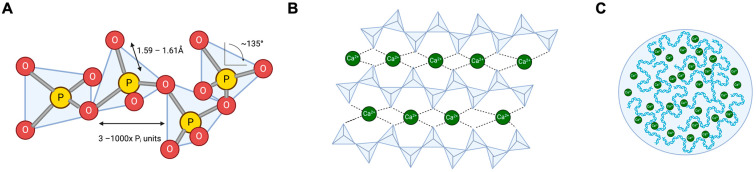
Gross structures of natural polyP. (**A**): PolyP is a polymer consisting of 3–1000 tetrahedral phosphate subunits linked to each other via a shared oxygen atom. Most polyP is linear; however, branched forms exist. (**B**): PolyP binds Ca^2+^, which binds further polyP and lines up, presumably in a helical secondary structure [27]. (**C**): Aggregated Ca^2+^ polyP forms nanoparticles that, in turn, form microsomes within cells.

**Table 1 biomolecules-14-00937-t001:** Major enzymes involved in metabolisms of polyP in different contemporary species.

Enzyme	Function	Regulation	Species	References
PPK1	polyP polymerase using ATP		Bacteria and *Dictyostelium discoideum* in cytoplasm	[39,40]
PPK2	polyP polymerase using GTP/ATP		Bacteria in cytoplasm	[41]
VTC	polyP polymerase using ATP	V-ATPase	*S. cerevisiae*, Trypanosoma and Leishmania in acidocalcisomes	[42,43,44]
PPN1	Endopolyphosphatase		*S. cerevisiae*, mammals	[45,46]
PPX1/2	Exopolyphosphatase	Regulated by the accumulation of polyP in the cytosol	Fungi, *S. cerevisiae* in mitochondria	[25,47,48,49,50]
Nudt (Nudix hydrolase 15)	PolyP hydrolase	Regulates oxidative stress in response to polyP	Mammals	[51]
Prune/DRES17	Short-chain exopolyphosphatase	Regulated by cytosolic polyP-level	Mammalian and *Drosophila melanogaster* cells in the cytosol	[49,52,53,54]
DDP1 (Diphosphoinositol polyP phosphohydrolase)	Endopolyphosphatase		*S. cerevisiae* in the cytosol	[45]
Vtc4 (Vacuolar transporter chaperone-4)	polyP polymerase	not specified	*S. cerevisiae* in Vacuole	[42,55]
PPN2	Endopolyphosphatase, Metallophosphatase	Zn^2+^	*S. cerevisiae* in acidocalcisomes	[56]
PPX2	Exopolyphosphatase	Mg^2+^, Mn^2+^, and Ca^2+^	bacteria	[57]

**Table 2 biomolecules-14-00937-t002:** Gross functions of extracellular and intracellular polyP.

Extracellular Functions	Intracellular Functions
Regulation of cell motility	Phosphate and energy storage
Contact activation of Factor XII and, consecutively, the kallikrein–kinin system and the intrinsic coagulation pathway	Regulation of Ca^2+^ homeostasis and signalling
pH regulation	Activating of mTOR-dependent pathways
Cation binding	Regulation of growth, differentiation, and apoptosis
Biofilm formation	Complexation of toxic heavy metal ions
Storage of Ca^2+^ and other ions in the ECM	Regulation of mitochondria permeability
	Interference with oxidative stress
	pH regulation
	Osmoregulation
	Regulation of enzyme activities

**Table 3 biomolecules-14-00937-t003:** Methods for analysing polyphosphate.

Detection Method	Description	Sensitivity	Reference
Light- and fluorescence microscopy	The use of specific dyes such as DAPI/Nile-Red for labelling polyP granules in cells enables qualitative detection + localisation of polyP	+/++	[145]
Flow cytometry	Localisation and quantification of polyP granules; fluorescent dyes such as DAPI + Nile-Red; enables sorting of cells based on polyP content	++/+++	[146,147]
Fluorescence in-situ hybridisation (FISH)	Identification of PAOs by specifically labelled probes for polyP formation or degradation genes enables the detection of DNA or RNA molecules in situ	++	[147,148]
Extraction procedures and polyphosphate quantification	Extraction of polyP from samples; measurement of phosphate concentration after hydrolysis of polyP; characterisation of different polyP fractions; quantification after enzymatic treatments	++/+++	[149]
UREA polyacrylamide gel electrophoresis (UREA-PAGE)	Separation of polyP molecules based on their size; determination of the degree of polymerisation (DP); use of specific staining techniques	+/++	[150]
Electron microscopy (EM)	Visual detection of polyP granules within cells; use of contrast agents for labelling and visualisation of polyP granules	+/++	[151,152]
X-Ray analysis (X-RAY)	Determination of the chemical composition of polyP granules; possible quantitative analysis; use of X-rays	++/+++	[153]
Nuclear magnetic resonance spectroscopy (NMRS)	Investigation of the structure of polyP and the phosphate metabolome; use of specific pulses for the detection of phosphorus atoms	++/+++	[154]
RAMAN spectroscopy	Identification and quantification of polyP in bacterial cells using characteristic Raman bands; use of specific spectra to recognise and quantify polyP	++/+++	[154,155,156]
Enzyme assays	PolyP-driven ATP formation is quantified by luciferase reporters	++	[157,158]
Cryo-electron tomography and spectroscopic imaging (CTSI)	Investigation of the structure of polyP granules in bacterial cells; high-resolution visualisation without destroying the sample; use of specific contrast agents and illumination techniques	+/++	[159]
Mass spectrometry (MS)	Characterisation of different forms of polyP molecules; identification and quantification of polyP molecules using specific ionisation and detection techniques	++/+++	[2]
Protein affinity	Detection and localisation of polyP in cells using specific polyP probes (PPBD)	++/+++	[143]
Omics’ technology (OMICS)	Global detection of polyP; identification of polyanions by specifically labelled probes/primers	++/+++	[160,161,162]

+ low sensitivity; ++ medium sensitivity; +++ high sensitivity; +/++ between low and medium sensitivity; ++/+++ between medium and high sensitivity.

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
