# Peer review of "An Update on Polyphosphate In Vivo Activities"

_biomolecules, 2024, doi:10.3390/biom14080937_

Round 1

Reviewer 1 Report

Comments and Suggestions for Authors

Overall, I found the review to be quite in-depth regarding the relevant pathways in which polyP is implicated, such as the plasma contact system and clotting. The review also highlights gaps in our knowledge, including polyP's role in nervous system pathways and the unknown human source of polyP.

As an update on polyP, perhaps more emphasis should be placed on the latest developments in polyP research. Given authors’ expertise in the polyP/coagulation field, it is understandable that the review focuses heavily on blood clotting and related aspects. However, new advances, such as the discovery of polyP interactions with polyHis and polyLys, could explain how many His-rich and Lys-rich proteins exert their polyP-related functions. Additionally, the role of His-rich glycoprotein in coagulation has garnered some attention.

As a review paper, it would be useful to include a final section on Future Perspectives.

In summary, I fully support the publication of this review. Here are some minor points to consider:

  • In the detection methods section, the authors mention the JC-D7 probe and other probes but could also include the FITC probe.
  • In some bacterial species, there is more than one type of PPK2, which should be briefly mentioned.
  • The writing seems a bit casual. There are many typos and minor issues; the authors should thoroughly review the entire manuscript.

Line 32  [Zitate]??

Line 39  enzymes, that regulation of polyP in – that??

Line 44  produced technically – technically??

Line 85 B: PolyP binds Ca2+, which in turn bind further polyP and line up, 85 presumably in a helical secondary structure – any reference to back this up?

Line 113 Is established for more than 60 – Is at the beginning of the paragraph??

Line 143 are positive for polyP – positive?

Line 178 greatly varies – vary

Line 179 highest in the nucleus (89 ± 7 μM) > plasma membrane – sentence is problematic

Line 185 has remained – have

Line 189 so far knwon has no polyP modifying activity - so far has no known polyP modifying activity

Line 235 has been identified – have been identified

Line 239 and in contrast to intracellular acidocalcisomes – remove and

Line 241 period missing after [18]

Line 327 In intracellular – remove In

Line 350 histidine-rich clusters – His-polyP literature says it is not covalent

Line 351 a novel covalent – a non-covalent

Comments on the Quality of English Language

I listed some minor issues, although far from exhaustive 

Author Response

Reviewer 1 - please also see attachment

Overall, I found the review to be quite in-depth regarding the relevant pathways in which polyP is implicated, such as the plasma contact system and clotting. The review also highlights gaps in our knowledge, including polyP's role in nervous system pathways and the unknown human source of polyP.

We are grateful to the reviewer for constructive remarks and time spent in reviewing our work.

As an update on polyP, perhaps more emphasis should be placed on the latest developments in polyP research. Given authors’ expertise in the polyP/coagulation field, it is understandable that the review focuses heavily on blood clotting and related aspects. However, new advances, such as the discovery of polyP interactions with polyHis and polyLys, could explain how many His-rich and Lys-rich proteins exert their polyP-related functions.

Lysine and histidine residues are positively charged at physiological pH and these amino acids bind tightly electrostatically to the negatively charged polyP backbone. Conserved histidine α-helical domain (CHAD) specifically binds to polyP and is found in a number of procaryotic cells [1]. However, charge seems not the only determinant for polyP binding as mostly lysine and histidine rather than positively charged arginine residues are important for polyP interaction. Furthermore, other negatively charged linear polymers such as single-stranded DNA and heparin differ from polyP in binding to proteins with positive charge clusters [2, 3]. We added information on CHAD to the review (lines 374-378).

Additionally, the role of His-rich glycoprotein in coagulation has garnered some attention.

The blood circulating histidine-rich glycoprotein (HRG) binds to polyanions such as DNA, RNA and polyP. HRG has been shown to inhibit polyP procoagulant activity  by interfering with FXII contact activation [4].  The underlying mechanism for antagonizing FXII activity is a competition of HRG with FXII for binding to polyP. Reduced FXII binding to polyP results in defective FXII contact activation and aPTT prolongation [5, 6]. Consistently, polyP-injection in HRG-deficient (Hrg-/-) mice triggers excess fibrin formation. [7, 8]. Vice versa reconstitution of Hrg-/- mice with HRG restores excess polyP driven thrombosis in a FXII dependent manner [4]. Taken together, the data are consistent with HRG operating as endogenous FXII regulator by binding to the contact activator polyP. We added the role of HRG in coagulation to the manuscript (lines 332ff).

As a review paper, it would be useful to include a final section on Future Perspectives.

As suggested, we added some sentences describing the most urgent needs for the field e.g. A polyP atlas that describes the amount and chain-length in various mammalian organs and as a next step in all cellular sub-compartments. Furthermore, the polyP metabolizing enzymes (if existing) need to be identified and fully characterized to allow for selective and specific modulation of polyP content, chain-length and thus activities. We added some thought to the end of the review (lines 408-414).

In summary, I fully support the publication of this review. Here are some minor points to consider:

In the detection methods section, the authors mention the JC-D7 probe and other probes but could also include the FITC probe.

Elegant work using FITC-labeled polyP has shown that cells endocytose polyP microparticles. Using the technology allows for intracellular polyP staining and analyses of transport and enrichment of (labeled) polyP to distinct cellular compartments depending on cell activation, metabolic state and environment [9]. We referenced FITC-labeled polyP and added the information to the manuscript (lines 393-397).

In some bacterial species, there is more than one type of PPK2, which should be briefly mentioned.

Corynebacterium glutamicum expressed two PPK2 isoenzymes while Pseudomonas aeruginosa and Sinorhizobium meliloti have three isoenzymes, each and Ralstonia eutropha even expresses five PPK2 variants [10]. Class I PPK2´s phosphorylate nucleoside diphosphates ADP and GDP by transferring the terminal Pi of polyP to the nucleoside diphosphates and vice versa shuttling Pi to the nascent polyP chain [11]. Class II PPK2´s catalyse nucleoside monophosphate phosphorylation and were previously referred to as polyP-AMP phosphotransferase (PAP) [12]. Class III PPK2´s phosphorylate nucleoside mono- or diphosphates generating ADP or ATP using polyP as phosphate donor [13]. Information was added to the review (lines 130-137).

The writing seems a bit casual. There are many typos and minor issues; the authors should thoroughly review the entire manuscript.

 Thank you for attention in detail. We carefully revised and corrected our entire manuscript

Line 32 [Zitate]??

Corrected we added [14].

Line 39 enzymes, that regulation of polyP in – that??

We corrected the sentence to “that regulate polyP in mammalians and in humans.” in line 39

Line 44 produced technically – technically??  Synthetically?

We replaced “technically” by “synthetically” in line 44.

Line 85 B: PolyP binds Ca2+, which in turn bind further polyP and line up, 85 presumably in a helical secondary structure – any reference to back this up?

We added a reference for Ca2+-mediated assembly of polyP  [15] (line 86).

Line 113 Is established for more than 60 – Is at the beginning of the paragraph??

We changed the sentence to “It is established for more…” (in line 113)

Line 143 are positive for polyP – positive?

We changed the sentence to “as the model system Saccharomyces cerevisiae contain polyP.” (line 149)

Line 178 greatly varies – vary

Corrected to “vary” in line 185.

Line 179 highest in the nucleus (89 ± 7 μM) > plasma membrane – sentence is problematic

We added the original reference and the classical methodology 185-188.

Line 185 has remained – have

We changed “has remained” to “have remained” in line 192.

Line 189 so far knwon has no polyP modifying activity - so far has no known polyP modifying activity

Corrected. Line 196.

Line 235 has been identified – have been identified

We have changed “has been identified” to “have been identified” in line 247.

Line 239 and in contrast to intracellular acidocalcisomes – remove and

We removed “and” in line 251.

Line 241 period missing after [18]

Corrected (line 256).

Line 327 In intracellular – remove In

We removed the “in” in line 349

Line 350 histidine-rich clusters – His-polyP literature says it is not covalent

We changed the sentence to “hypothesis of polyP being a novel non-covalently attached posttranslational modification” in lines 369ff.

Line 351 a novel covalent – a non-covalent

Corrected! We changed the sentence to “hypothesis of polyP being a novel non-covalently attached posttranslational modification” in lines 369ff.

References

  1. Lorenzo-Orts, L., et al., Molecular characterization of CHAD domains as inorganic polyphosphate-binding modules. Life Sci Alliance, 2019. 2(3).
  2. Neville, N., et al., Modification of histidine repeat proteins by inorganic polyphosphate. Cell Rep, 2023. 42(9): p. 113082.
  3. Neville, N., et al., Polyphosphate attachment to lysine repeats is a non-covalent protein modification. Mol Cell, 2024. 84(9): p. 1802-1810 e4.
  4. Malik, R.A., et al., Polyphosphate-induced thrombosis in mice is factor XII dependent and is attenuated by histidine-rich glycoprotein.Blood Adv, 2021. 5(18): p. 3540-3551.
  5. MacQuarrie, J.L., et al., Histidine-rich glycoprotein binds factor XIIa with high affinity and inhibits contact-initiated coagulation.Blood, 2011. 117(15): p. 4134-41.
  6. Malik, R.A., et al., Histidine-rich glycoprotein attenuates catheter thrombosis. Blood Adv, 2023. 7(18): p. 5651-5660.
  7. Banno, F., et al., Exacerbated venous thromboembolism in mice carrying a protein S K196E mutation. Blood, 2015. 126(19): p. 2247-53.
  8. Zilberman-Rudenko, J., et al., Factor XII Activation Promotes Platelet Consumption in the Presence of Bacterial-Type Long-Chain Polyphosphate In Vitro and In Vivo. Arterioscler Thromb Vasc Biol, 2018. 38(8): p. 1748-1760.
  9. Fernandes-Cunha, G.M., et al., Delivery of Inorganic Polyphosphate into Cells Using Amphipathic Oligocarbonate Transporters. Acs Central Science, 2018. 4(10): p. 1394-1402.
  10. Neville, N., N. Roberge, and Z. Jia, Polyphosphate Kinase 2 (PPK2) Enzymes: Structure, Function, and Roles in Bacterial Physiology and Virulence. Int J Mol Sci, 2022. 23(2).
  11. Racki, L.R., et al., Polyphosphate granule biogenesis is temporally and functionally tied to cell cycle exit during starvation in.Proceedings of the Nat Acad Sci U S A, 2017. 114(12): p. E2440-E2449.
  12. Bonting, C.F.C., G.J.J. Kortstee, and A.J.B. Zehnder, Properties of Polyphosphate - Amp Phosphotransferase of Acinetobacter Strain-210a. J Bacteriol, 1991. 173(20): p. 6484-6488.
  13. Motomura, K., et al., A New Subfamily of Polyphosphate Kinase 2 (Class III PPK2) Catalyzes both Nucleoside Monophosphate Phosphorylation and Nucleoside Diphosphate Phosphorylation. Appl Environment Microbiol, 2014. 80(8): p. 2602-2608.
  14. Kumar, A., et al., Polyphosphate and associated enzymes as global regulators of stress response and virulence in. World J Gastroenterol, 2016. 22(33): p. 7402-7414.
  15. Jackson, L.E., et al., Synthesis and structure of a calcium polyphosphate with a unique criss-cross arrangement of helical phosphate chains. Chem Mat, 2005. 17(18): p. 4642-4646.

Reviewer 2 Report

Comments and Suggestions for Authors

This manuscript presents an updated review on the functions of polyphosphates in various organisms from bacteria to higher mammalians. Overall the information presented is correct, although the inclusion of references to previous reviews focusing on the functions of polyPs in the microbial world is lacking, and there are errors about the ascription of some polyphosphatases to protein families and their evolutionary relationships that need to be corrected.

line 32: please, include citations on polyP roles in microorganisms ([zitate]?)

line 132:  respectively, both being members of the same protein family, the PPX-GppA phosphatases

line 141: consider replacing "expression" with "presence" or "occurrence"

line 176: consider deleting "Higher"

lines 183 and 186: consider replacing "lower" by "simpler"

lines 189-191:  The sentence is misleading and requires rewriting. Bacterial PPX1 and human (h)-Prune are functionally equivalent but non-homologous proteins, as they do not share a common ancestor. The former is a member of the family of PPX-GppA phosphatases, while the latter belongs to the superfamily of DHH phosphoesterases.

Table 1. The fifth row "PPX1/2" should be divided in two, one row for bacterial exopolpyphosphatase PPX1 (a member of the PPX-GppA family of phosphatases) and the other for fungal exopolpyphosphatases PPX1 and 2 (DHH-DHHA2 phosphoesterases)

line 300: should read "...(PPX), a member of the PPX-GppA phosphatases family, specifically..."

Table 3: should read "Sensitivity"; the fourth routh, first column shoud read "....Quantification after Enzymatic Treatments"; the sixth routh, first column shoud read "Electronic...."

line 388: check the last page of the reference 4.

Comments on the Quality of English Language

lines 55-58- Please, check and rewrite this long sentence; should read "solutions"

line 113: should read "polyP is produced"

Author Response

Reviewer 2

This manuscript presents an updated review on the functions of polyphosphates in various organisms from bacteria to higher mammalians. Overall the information presented is correct, although the inclusion of references to previous reviews focusing on the functions of polyPs in the microbial world is lacking, and there are errors about the ascription of some polyphosphatases to protein families and their evolutionary relationships that need to be corrected.

We thank you the time to reviewing our review and are grateful for your constructive comments. We will respond to your comments and will supplement and, if necessary, correct our work.

line 32: please, include citations on polyP roles in microorganisms ([zitate]?)

Added as suggested; reference [1] (reference 3 in the review, line 32).

line 132:  respectively, both being members of the same protein family, the PPX-GppA phosphatases

We added “both being members of the same protein family, the PPX-GppA phosphatases” in line 140

line 141: consider replacing "expression" with "presence" or "occurrence"

We replaced “expression” by “presence” in line 149.

line 176: consider deleting "Higher"

Thank you, we deleted “higher” throughout the manuscript.

lines 183 and 186: consider replacing "lower" by "simpler”

We replaced “lower” by “simpler” in line 191 and 194

lines 189-191:  The sentence is misleading and requires rewriting. Bacterial PPX1 and human (h)-Prune are functionally equivalent but non-homologous proteins, as they do not share a common ancestor. The former is a member of the family of PPX-GppA phosphatases, while the latter belongs to the superfamily of DHH phosphoesterases.

As suggested, we replaced the (now 196-200): Bacterial PPX1 and human (h)-Prune are functionally equivalent but non-homologous proteins originating from distinct ancestors. PPX1 is a member of the PPX-GppA phosphatases family, while h-Prune belongs to the of DHH phosphoesterases superfamily.

Table 1. The fifth row "PPX1/2" should be divided in two, one row for bacterial exopolpyphosphatase PPX1 (a member of the PPX-GppA family of phosphatases) and the other for fungal exopolpyphosphatases PPX1 and 2 (DHH-DHHA2 phosphoesterases)

As suggested, we divided the fifth row (originally “PPX1/2”) into “PPX1” and “PPX2”. “PPX1 (PPX-GppA family) – Exopolyphosphatase – regulated by polyP accumulation in the cytosol – bacteria” and “PPX1/2 (DHH-DHHA2 phosphoesterases) – Exopolyphosphatase- regulated by polyP accumulation in the cytosol – fungi” (Table 1)

line 300: should read "... (PPX), a member of the PPX-GppA phosphatases family, specifically..."

We added “a member of the PPX-GppA phosphatases family” to line 311.

Table 3: should read "Sensitivity"; the fourth routh, first column shoud read "....Quantification after Enzymatic Treatments"; the sixth routh, first column shoud read "Electronic...."

We changed “sensi-tivity” to “Sensitivity”, “use of specific enzymes for phosphate quantification” to “Quantification after enzymatic treatments” and “Electronen microscopy (EM)” to “Electron microscopy (EM)” (Table 3)

line 388: check the last page of the reference 4.

Corrected. (now ref. 5, p. 367-371.)

References

  1. Kumar, A., et al., Polyphosphate and associated enzymes as global regulators of stress response and virulence in. World J Gastroenterol, 2016. 22(33): p. 7402-7414.

Round 2

Reviewer 2 Report

Comments and Suggestions for Authors

line 32. The title of reference 3 seems incomplete. Please, check it. On the other hand, this reference is very specific and does not seem appropriate in the context of a sentence referring to micro-organisms in general. It is therefore mandatory to cite in this context a review with a broad view of the relevance of polyPs for the microbial world.

Otherwise, the article has been edited and modified suitably.

Comments on the Quality of English Language

-----

Author Response

line 32. The title of reference 3 seems incomplete. Please, check it. On the other hand, this reference is very specific and does not seem appropriate in the context of a sentence referring to micro-organisms in general. It is therefore mandatory to cite in this context a review with a broad view of the relevance of polyPs for the microbial world.

Otherwise, the article has been edited and modified suitably.

Thank you very much for the positive review and for your comment. As advised, we have exchanged the reference with two general reviews that comprehensively cover the role of polyP in bacteria.

The revised introduction now has:

[1, 2];  (References 3 and 4 in our review)

  1. Moreno, S.N. and R. Docampo, Polyphosphate and its diverse functions in host cells and pathogens.PLoS Pathog, 2013. 9(5): p. e1003230.
  2. Wang, L., et al., Distribution Patterns of Polyphosphate Metabolism Pathway and Its Relationships With Bacterial Durability and Virulence. Front Microbiol, 2018. 9: p. 782.